# Developmental Profile of Sleep and Its Potential Impact on Daytime Functioning from Childhood to Adulthood in Sickle Cell Anaemia

**DOI:** 10.3390/brainsci10120981

**Published:** 2020-12-14

**Authors:** Melanie Kölbel, Fenella J. Kirkham, Dagmara Dimitriou

**Affiliations:** 1Department of Developmental Neurosciences Unit, UCL Great Ormond Street Institute of Child Health, London WC1N 1EH, UK; melanie.koelbel.15@ucl.ac.uk; 2Clinical and Experimental Sciences, University of Southampton, Southampton SO17 1BJ, UK; 3Child Health, University Hospital Southampton, Southampton SO16 6YD, UK; 4Paediatric Neurosciences, King’s College Hospital, London SE5 9RS, UK; 5Sleep Education and Research Laboratory, UCL Institute of Education, 25 Woburn Square, London WC1H 0AA, UK; d.dimitriou@ucl.ac.uk

**Keywords:** blood disorders, sickle cell anaemia, sleep characteristics, obstructive sleep apnoea, polysomnography, developmental trajectory

## Abstract

Young individuals with sickle cell anaemia (SCA) experience sleep disturbances and often experience daytime tiredness, which in turn may impact on their daytime functioning and academic attainment, but there are few longitudinal data. Methods: Data on sleep habits and behaviour were taken on the same day as an in-hospital polysomnography. This study assesses the developmental sleep profiles of children and young adults aged 4–23 years old with SCA. We examined retrospective polysomnography (PSG) and questionnaire data. Results: A total of 256 children with a median age of 10.67 years (130 male) were recruited and 179 returned for PSG 1.80–6.72 years later. Later bedtimes and a decrease in total sleep time (TST) were observed. Sleep disturbances, e.g., parasomnias and night waking, were highest in preschool children and young adults at their first visit. Participants with lower sleep quality, more movement during the night and increased night waking experienced daytime sleepiness, potentially an indicator of lower daytime functioning. Factors influencing sleep quantity included age, hydroxyurea prescription, mean overnight oxygen saturation, sleep onset latency, periodic limb movement, socioeconomic status and night waking. Conclusion: Sleep serves an important role for daytime functioning in SCA; hence, quantitative (i.e., PSG for clinical symptoms, e.g., sleep-disordered breathing, nocturnal limb movement) and qualitative (i.e., questionnaires for habitual sleep behaviour) assessments of sleep should be mutually considered to guide interventions.

## 1. Introduction

The healthy human red blood cell is a biconcave disc, making it easy to move smoothly around the human vascular system, important for the continuous transport by haemoglobin of oxygen, essential for cellular respiration. Sickle cell anaemia (SCA) is a single-gene blood disorder that, in homozygotes, causes the red blood cells to become sickle shaped secondary to polymerisation of the abnormal sickle haemoglobin under hypoxic conditions. Polymerised haemoglobin also carries less oxygen and the sickled cell may stick to and block blood vessels [1]. Chronic hypoxia on exposure to sleep-disordered breathing (SDB), e.g., obstructive sleep apnoea (OSA), might increase the risk for structural and functional brain abnormalities leading to poor cognitive function [2,3,4,5,6]. The prevalence of SDB diagnosed with polysomnography (PSG) in SCA is high, ranging from 36% to 69% [7,8,9], due mainly to hyperplasia of adenoids and tonsils causing upper-airway obstruction [10], particularly in young children, although the anatomy of bony structures may also play a part [11]. Habitual snoring in children with SCA is considered as a risk factor for OSA [12] and >50% of children snore regularly [13], while >70% of adults with SCA reported sleep disturbances, which correlated with symptoms of depression [14]. In the general adult population, OSA is associated with lower quality of life, daytime sleepiness and poorer mental and physical health [15]. Other comorbid sleep disorders reported in SCA are periodic limb movement syndrome (PLMS), restless leg syndrome (RLS) and nocturnal enuresis [16,17,18]. Studies using questionnaires and sleep diaries have identified that 37.5% of 8–12-year-old children with SCA had sleep onset difficulties [19], 50% of 2–18 year olds had delayed sleep onset insomnia and 21% experienced chronic insomnias [13]. In the general paediatric population, 11.3% of 5–12 year olds (85.7% white) experienced sleep onset insomnia and 6.5% wake up frequently at night [20], while 10.7% of 13–16 year olds (62.3% white) had chronic insomnia [21]. Information provided directly from daily sleep diaries of 8–12-year-old children with SCA indicated that 66.7% experienced nocturnal night waking [19]. In contrast, their parents only reported 29.1% of their children to frequently wake up during the night. Research in the general population has compared parent-reported sleep with either actigraphy or PSG. For children aged 4–9 years, parents overestimated sleep duration by an average of 24 min compared with actigraphy measures [22]. A recent study has compared parent and child sleep report of 9–17-year-old youth to their child’s PSG data [23]. Parent and child overestimated measures of total sleep time (TST), sleep latency and sleep efficiency compared to PSG measures. The majority of studies include relatively small cohorts from different racial backgrounds, with the literature either missing sleep characteristics data in young children of black heritage or exclusively examining their sleep characteristics. Caretaker reports of 2–8-year-old children of black heritage showed overall lower nocturnal sleep duration, with 20 min less sleep on weekdays compared to white children [24]. An actigraphy study in 14–19-year-old adolescents (57% black) identified a shorter nocturnal sleep duration in black females (6.3 h) compared to white females (6.9 h) [25]. The same study identified black males to have higher sleep fragmentation (32.5) compared to white males (26.9). This is in line with a recent review indicating that black adults tend to experience shorter sleep duration, lower sleep quality and daytime sleepiness compared to non-Hispanic White adults [26]. Racial differences in sleep behaviour may be mediated by sociodemographic, level of education, household income and pre-existing health and sleep disorders [26,27]. Several studies have also shown that prolonged disturbances, such as reduced sleep quality, often result in daytime sleepiness, lowering receptivity at work and school [28]. This contributes to a large proportion of problems with academic attainment [29], cognitive development and impairment [30], as well as mental health problems (e.g., depressive symptoms) [31]. For example, the chronotype of an individual, such as late sleepers, show an increased risk of cardiovascular disease in adults [32]. A recent study showed that children (aged 6–12 years) with an evening chronotype experience more sleep disturbances and depressive symptoms [33]. Fatigue is common in the anaemic SCA population, present in 69% of 15–30 year olds [34], which is shown to be influenced by sleep quality [35] and thus, could aggravate clinical symptoms and lower quality of life. Despite the high prevalence of symptomatic sleep disorders in SCA, the pathophysiology and natural history are poorly understood, in part because the majority of polysomnography (PSG) studies have recruited mainly those with symptoms of sleep disorders. There are cross-sectional studies involving questionnaires, but few have compared with contemporaneous PSG or included control groups and no research has gathered longitudinal data on sleep behaviour in SCA. To our current knowledge, there are only two cross-sectional studies available that have used PSG with age, gender, and ethnicity matched controls to compare sleep behaviour in SCA [36,37]. Both found a non-significant lower TST in children with SCA compared to controls, with one study showing a 17 min difference [36] and the other a 24 min difference [37]. However, one actigraphy study found a significant difference in adolescents with SCA (aged 12–17 years) compared to ethnicity-matched controls, with SCA sleeping 1 h and 15 min longer [38]. There appears to be a discrepancy between PSG, actigraphy and questionnaire data, with a contrast between in-hospital PSG studies and questionnaires and actigraphy reflecting the situation at home. The purpose of this retrospective analysis of a prospective observational longitudinal study of polysomnography and sleep questionnaires was to examine sleep characteristics and developmental sleep profiles in children and adolescents with SCA not selected for a history of sleep disorder. We also aimed to compare PSG data to self-reported measures to identify differences and determine whether both measures equally show a decline in TST longitudinally, with their implications for daytime functioning and health in SCA.

## 2. Materials and Methods

### 2.1. Participant Selection and Description

For the London site, ethical approval was granted by the Brent (05/Q0408/42) and East Midlands-Leicester (11/EM/0084) National Research Ethics Service Committees; and for the US sites, approval was gained from the Institutional Review Boards of Washington University School of Medicine, St. Louis, MO, Federal Fide Assurance (FWA) #00000163 and Case Western Reserve, Cleveland, OH (FWA #00004428). Details for ethical approval, recruitment and follow up of children and young adults with SCA to the Sleep Asthma Cohort (SAC), investigating the prevalence and importance of sleep disturbances and asthma, have been previously published [12,39,40]. Each child and adolescent with SCA aged 4–18 attending their routine hospital visits was eligible to take part regardless of past morbidity or sleep-disordered breathing symptoms. The exclusion criteria were: co-occurring chronic lung disease other than asthma, respiratory failure, additional respiratory support at night and regular long-term blood transfusions. Written informed consent was either taken directly from the participant and/or parent/guardian. Where appropriate participant data were grouped into: (1) preschool (4–5.99 years); (2) school-aged children (6–13 years); (3) teenagers (14–17 years); and (4) young adults (18–24 years) according to the National Sleep Foundation’s sleep quality recommendations [36].

### 2.2. Procedures

After taking consent from either the parent/guardian and/or their child(ren), we asked participant and/or parent/guardian to answer questions about their own or their child’s sleep behaviour and habits, as well as to fill in demographic questions. After a short break, the participant was prepared to undergo their polysomnography in-hospital assessment.

#### 2.2.1. Polysomnography

The equipment (in-laboratory PSG; Embla N-7000) and PSG measures obtained for SAC are described in detail elsewhere [12,41,42]. PSG data were recorded from the time of the habitual bedtime and ended as soon as the child woke up spontaneously or at the latest by 7 a.m. [41]. Study certified technicians followed the American Academy of Sleep Medicine guidelines for PSG scoring and standardised protocols. Since sleep profiles might differ, PSG recordings were classified as weekday (i.e., Sunday to Thursday night) and weekend recordings (i.e., Friday–Saturday night) for comparison with parent-reported general sleep characteristics.

#### 2.2.2. Sleep Characteristics and Demographics Questionnaire

Parents/guardians completed standardised questionnaires about their child’s sleep behaviour. An adapted version of the Children’s Sleep Habits Questionnaire (CSHQ) [43] was used; the baseline data have been published [12,44]. Parents rated their child’s sleep habits on a frequency scale: never (does not happen), not often (1 night/day per week), sometimes (1–2 nights/days per week), often (3–5 nights/days per week), and always (6–7 nights/days per week). A score was given to assess sleep problems during the last month with 0 points (never) to 4 points (always: 6–7 night/s week). Included variables were clinical symptoms (i.e., snoring, limb movement and fatigue), general sleep/wake times during week/weekend and day-time behaviour (Appendix A). A sleep composite score was calculated from all combined 20 questions relating to sleep problems at night based on previous research [44]. Higher sleep composite score was indicative of lower sleep quality. Indication of daytime functioning (i.e., sleepiness) was assessed with the question “Does your child act sleepy or overtired a lot during the day?” The questions followed the same frequency scale as the CSHQ. A score for napping behaviour was created (i.e., 0 = never; 1= not often; 2= sometimes, 3 = often, 4 = always). Information on parental marital status, household income and head of household education was obtained to provide an index of socioeconomic status (SES). An indication of chronotype was calculated for the midpoint of sleep using reported bedtime and waketime.

### 2.3. Statistical Analysis

Statistical analysis was performed with SPPS version 26. Test of normality and homogeneity of variance were checked with the Shapiro–Wilk test. CSHQ and PSG were compared with the Wilcoxon’s signed rank test, given the non-normality of variables. Cook’s distance was used to identify potential outliers. Developmental trajectory analysis using repeated measures ANOVA was used to assess relationships between sleep measures and chronological age (CA) [45]. Differences in age groups were assessed using ANOVA or the Kruskal–Wallis test. Furthermore, exploratory hierarchical multiple regression analysis was used to look at predictor variables of the sleep variables. Bivariate correlational analysis helped to determine the inclusion and exclusion of variables in the regression analysis. Assumptions were checked for singularity, multicollinearity, and residuals using appropriate statistics and scatter plots. Factors were entered into the model based on the previous studies within this research area [16].

## 3. Results

### 3.1. Recruitment and Baseline Characteristics

For the first sleep study (see, Figure 1) (Visit 1, thereafter V1), 256 individuals (*N* = 130, 51% male) were recruited with a median CA of 10.67 (range: 4.00–23.8 years). In the second follow-up visit (Visit 2, thereafter V2), 204 individuals (102, 50% male) returned to complete questionnaires at a median CA of 14.73 (range: 6.43–24.98 years). However, a smaller number of 179 (89, 48% male) took part in the follow-up PSG at a median CA of 14.66 (range: 6.43–23.95 years; 86 male). The attrition rate was 20% for return for the questionnaires and 30% (*N* = 77) for the PSG, which is lower than expected, or consistent with expected, respectively, for this clinical sample, estimated at approximately 30%. The main reasons for not attending the second visit was time commitment, personal circumstances (i.e., sickle crises, moving home and admission to hospital) and discomfort of PSG in the hospital. The average time difference ±SD between both sleep studies was 4.53 (1.08) years, with a range from 1.80 to 6.72 years.

Demographic characteristics of the participants are described in Table 1. There was no significant difference between participants taking part in the follow-up PSG, and those who did not return (i.e., who did not have a second in-hospital PSG), in CA, gender, use of hydroxyurea, household income and head of household education (all *p* > 0.05). Parental marital status was significantly different (*p* > 0.001) (Table 1). Data analyses of participants who did not return for V2 visit showed that they were less symptomatic compared to those returning for their second PSG night (Table 1).

### 3.2. Comparison of Sleep Profiles as Measured with PSG and CSHQ

Sleep data collected from PSG and CSHQ showed a significant change from V1 to V2 on both measures. When compared to one another (i.e., PSG V1 vs. CSHQ V1), significant differences were observed. Mean differences (±SD) for both measures are shown in Table 2. Later bedtime and a decrease in total sleep time (TST) were observed for both measures from V1 to V2. Detailed description of sleep characteristics for all age groups from V1 to V2 are shown in Appendix A for CSHQ and Appendix A for PSG. During their PSG night, younger children had a later than usual bedtime (BT) (i.e., measured with CSHQ) (e.g., school-aged children at V1:Mean_CSHQ_ = 21:36 (00:49) vs. Mean_PSG_ = 22:05 (00:49)) than young adults, who showed to have an earlier bed time during their overnight PSG (e.g., young adults at V1:Mean_CSHQ_ = 00:09 (01:20) vs. Mean_PSG_ = 22:40 (00:46)). Wake time (WT) during PSG was earlier than usual for all participants and increased for CSHQ and decreased for PSG at V2. CSHQ has shown that young adults tend to go to bed between 09:30 pm and 3:00 am on weekdays and wake up later than their younger peers (e.g., school-aged children at V1:Mean_CSHQ_ = 06:54 am (00:45) vs. young adults V1:Mean_CSHQ_ = 07:40 am (01:46) and V2: 08:40 am (02:02). At the weekend, young adults tend to go to bed between 10:00 pm and 3:00 am and wake up later than their younger peers (e.g., school-aged children at V1:Mean_CSHQ_ = 09:07 (01:29) vs. young adults Mean_CSHQ_ = 10:02 (01:25). PSG indicated a decrease in sleep onset latency (SOL), while CSHQ reported values showed an increase in SOL at V2. Sleep efficiency, as measured with PSG, decreased with time (mean V1 = 92% (7.5) vs. mean V2 = 81% (11.8), *p* < 0.001.

Overall, 70% of participants took more than 20 min to fall asleep, shifting from school-aged children with the most problems observed at V1 and young adults at V2. During the weekdays, school-aged children showed higher SOL of up to 180 min at V1 and 210 min at V2, while adolescents showed higher sleep latency at the weekend of up to 180 min at V1 and 240 min at V2.

Gender differences were only observed for measures taken from CSHQ for different age categories. For example, V1 for preschool children: BT (mean ♀ = 21:45 (00:55) vs. mean ♂ = 21:04 (00:58)) and WT (mean ♀ = 08:24 (00:59) vs. mean ♂ = 07:45 (00:54)). Adolescents showed differences on the CSHQ V2 for TST (mean ♀ = 08:37 (01:25) vs. mean ♂ = 07:27 (01:30) and SOL during PSG V2 (mean ♀ = 00:32 (00:30) vs. mean ♂ = 00:16 (00:30)), all *p* < 0.05. According to the National Sleep Foundation’s (NSF) recommendation for sleep duration [46], 63% of preschool children slept less than the recommended amount of 10–13 h. At V2, it was recorded that 36% of school-aged children, 40% of teens and 15% of young adults slept less than the recommended amount (i.e., sleep recommendation for (1) school-aged children 9–11 h, (2) teens 8–10 h and (3) young adults 7–9 h). Data obtained from the CSHQ suggest that sleep need seems to increase at V2, since 11.4% of adolescents and 21% of young adults sleep more than the recommended by the NSF.

### 3.3. Developmental Sleep Profile Changes

Developmental sleep profile changes were observed for TST as taken from overnight PSG and CSHQ self-report data. Values are shown as mean (hh:mm) ± SE. PSG TST significantly decreased with age over time 00:46 (6.54), (95% CI, 00:33–00:59): *F* (1169) = 49.61, *p* < 0.001, partial *η2* = 0.23. There was a significant interaction between age difference and TST: *F* (1167) = 7.94, *p* = 0.005, *η2* = 0.05 (see, Figure 2a). CSHQ TST significantly decreased with age over time 00:39 (7.56), (95% CI, 00:24–00:54): *F* (1197) = 27.33, *p* < 0.001, partial *η2* = 0.12 (see, Figure 2b). CSHQ TST during weekdays significantly decreased with age over time 00:33 (7.72), (95% CI, 00:17–00:48): *F* (1198) = 18.52, *p* < 0.001, partial *η2* = 0.086 (see, Figure 2c). CSHQ TST during weekends significantly decreased with age over time 00:38 (9.93), (95% CI, 00:19–00:58): *F* (1197) = 15.12, *p* < 0.001, partial *η2* = 0.071 (see, Figure 2d). No significant interaction for all measures of CSHQ was observed between age difference and TST (all, *p* > 0.05). No gender difference in TST between male and female were observed for all measures.

### 3.4. Midpoint of Sleep

The midpoint of sleep, as calculated from the CSHQ, shifted further away from midnight (mean V1 = 03:00 (00:55), range = 01:07–07:00 vs. mean V2 = 03:42 (01:15), range = 01:30:00–08:15:00, *p* < 0.01), with school-aged children experiencing the earliest midpoints (mean = 03:01 (00:44), range = 01:30–05:30) followed by teens (mean = 03:50 (00:53), range = 02:11–06:45) and young adults the latest (mean = 04:56 (01:24), range = 02:00–08:15), *p* < 0.001.

During the week an early midpoint (i.e., 01:00–03:00) was shown in 90% (V1) and 75% (V2), while a late midpoint (i.e., 05:00–09:00) was shown for 1% (V1) and 8%(V2). During the weekend, an early midpoint (i.e., 01:00–03:00) was shown in 29% (V1) and 16% (V2), while a late midpoint (i.e., 05:00–09:00) was shown for 24% (V1) and 46%(V2). However, midpoint of sleep for PSG shifted slightly towards midnight at V2 (mean V1 = 02:38 (01:26), range = 23:50–01:09 vs. mean V2 = 02:28 (00:28), range = 01:09:00–04:30:00, *p* = 0.024).

### 3.5. PSG: Clinical Sleep Characteristics

As predicted, for PSG measures of obstructive sleep apnoea (OSA, measured with Apnoea and Hypopnea Index (AHI)) and periodic limb movement (measured with the periodic limb movement index (PLMI)), the V2 values were overall significantly lower than the V1 values (Table 3). No sex differences were observed for all PSG scores (*p* > 0.05) at V1 and V2. At their first visit, 5% had severe OSA according to the clinical cut off >10, while 9% of participants showed moderate OSA (≥5 and <10) and 43.4% showed mild OSA (≥1 and <4.9). A detailed comparison between the different age groups is shown in Appendix A. Children in the youngest age group showed the highest AHI mean score (±SD) of 3.94 (7.21) followed by school-aged children with AHI of 2.34 (4.25). A reduction in OSA symptoms was observed in both groups at V2, with 2.9% having severe symptoms, but a slight increase to 46.47% for mild symptoms. School-aged children showed a slight reduction in AHI (6.43–13.99 years) to 2.12 (3.05). At V1, 37.7% showed mild periodic limb movement (PLMI < 5 and >25) and only 0.4% had moderate (PLMI > 25 and <50), and 0.7% severe symptoms (PLMI > 50). An overall reduction in PLMI symptoms was observed during the second visit, but 6.51% showed now moderate symptoms. An increase in PLMI was observed for young adults from V1 (mean = 4.00 (5.26)) to V2 (mean = 5.19 (11.81)). WASO and night waking increased overall between V1 and V2, while mean overnight oxygen saturation decreased (Table 3).

### 3.6. CSHQ: General Sleep Characteristics

As predicted, there was a significant difference in CSHQ composite score (CSHQ CS) between V1 and V2 (Table 4). No differences were observed between gender in CSHQ CS and subscales (*p* > 0.05) at V1 and V2. During V1, 17.7% of the sample scored above the mean (±2SD) for CSHQ CS and were more likely to experience lower sleep quality with symptoms of parasomnias (15.7%), sleep-disordered breathing (20.8%), night waking (19.2%) and bedtime resistance (17.7%). Habitual snoring (i.e., between 3 and 7 times a week) was present in 23%. A detailed comparison between the different age groups is shown in Appendix A. Preschool children showed the highest CSHQ SC mean score (mean = 17.7 (9.43)), followed by young adults with 15.3 (±10.51). Similar mean CSHQ subscale scores for sleep-disordered breathing were observed for preschool children (mean = 3.19 (3.34)) and school-aged children (mean = 3.48 (3.23)). Parasomnias were highest in preschool children (mean = 6.90 (4.37)) and young adults (mean = 5.00 (5.20)). Night waking was highest in preschool children (mean = 4.48 (3.52)) and young adults (mean = 4.85 (3.98)). During V2, a lower CSHQ CS was observed, which could imply better sleep quality. However, 15.8% still scored above the mean (±2SD) for sleep-disordered breathing symptoms (19.8%), night waking (18.3%) and greater bedtime resistance (19.8%), but lower symptoms of parasomnias (12.4%) and increased sign of habitual snoring (i.e., between 3 and 7 times a week) was present in 28%. CSHQ CS mean score (±SD) decreased for school-aged children (mean = 15.31 (10.51)) and young adults (mean = 13.63 (8.21)) indicating better sleep quality. The same accounts for parasomnia symptoms and night waking.

### 3.7. Daytime Functioning

At V1, 36% suffered from daytime sleepiness and/or overtiredness at least once a week. However, sleepiness increased to 49% for SCA participants at V2. The heatmap shows the relationship for CSHQ scores, napping behaviour, TST and haemoglobin (Table 5). Participants with higher CSHQ CS, indicating lower sleep quality, experience sleepiness more often during the week. The same accounts for SDB, parasomnia and frequent night waking. Greater movement during the night, suggestive of periodic limb movement, is also associated with greater sleepiness in SCA participants, especially at V2. Usually, napping behaviour increases with increased occurrence of sleepiness as well. Interestingly, participants who experienced shorter TST, also felt less tired at V1, while a change occurred during V2 and participants who now slept the least felt always tired during the day. Participants with the lowest haemoglobin levels also felt sleepier throughout the day compared to those that had higher haemoglobin levels.

### 3.8. Predictors of Total Sleep Time 

The potential differences for participants receiving hydroxyurea treatment and relationships of TST were investigated. Values are shown as mean (hh:mm) ± SD. No differences were found for hydroxyurea use and TST at PSG V1 and V2 (*p* > 0.05.) However, CSHQ TST identified a significant difference in hydroxyurea use. Individuals taking hydroxyurea (*N*_V1_ = 31, *N*_V2_ = 25,) had lower TST at V1_TST_ 8:47 (1:26) and V2_TST_ 8:11 (1:30), compared to those that did not (*N*_V1_ = 203, *N*_V2_ = 163) at V1_TST_ 9:29 (1:16) and V2_TST_ 8:52 (1:27), (both, *p* < 0.05).

#### 3.8.1. Predictors of Total Sleep Time (PSG)

Age and gender were entered first into the model as control variables. The second step included disease variables hydroxyurea, AHI, mean overnight oxygen saturation (O_2_) and periodic limb movement (PLMI). The third step added socioeconomic (SES) variables near poverty level (<$20,000) and primary caregiver that is divorced/separated (marital status). The fourth step included sleep onset latency (SOL), wake after sleep onset (WASO), number of night waking and wake time in the morning. After careful consideration of the data and model assumptions, gender (step 1), hydroxyurea (step 2) and SES (step 3) were removed. None of the variable had an influence on the full model.

The hierarchical multiple regression model showed that age contributed significantly to the model at stage one, *F* (1, 251) = 7.39, *p* = 0.007, adjusted *R^2^* = 0.03. Introducing O_2_ and PLMI at stage 2 revealed that only PLMI was a significant predictor, *F* (3, 251) = 4.85, *p* = 0.003, adjusted *R*^2^ = 0.04. The final step revealed that the full model of age, O_2_, PLMI, SOL, WASO, night waking and wake time was predictive of total sleep time (PSG) at V1 and was statistically significant, *F* (7, 251) = 41.11, *p* < 0.001, adjusted *R*^2^ = 0.53 (see, Table 6). The model found that TST tends to decrease with: (a) older age, (b) higher mean overnight O_2_, (c) increase in periodic limb movement, (d) longer SOL and (e) longer periods of wakefulness during the night (WASO). TST tends to increase with less night waking and later waking time in the morning.

#### 3.8.2. Predictors of Total Sleep Time (CSHQ)

Age and gender were entered first into the model as control variables. The second step included disease variables hydroxyurea, AHI, mean overnight oxygen (O_2_) and periodic limb movement (PLMI). The third step added socioeconomic (SES) variables near poverty level (<$20,000) and primary caregiver that is divorced/separated (marital status). The fourth step included sleep onset latency (SOL) and wake time in the morning. After careful consideration of the data and model assumptions, marital status was removed (step 3). The hierarchical multiple regression model showed that age and not sex, contributed significantly to the model at stage one, *F* (2, 230) = 31.56, *p* < 0.001, adjusted *R*^2^ = 0.21. Introducing disease variables at stage 2 revealed that PLMI is a significant predictor of the model, with age remaining significant, *F* (6, 230) = 12.58, *p* < 0.001, adjusted *R*^2^ = 0.23. Adding SES to the model revealed a significant change to the model, *F* (7, 230) = 11.72, *p* < 0.001, adjusted *R*^2^ = 0.25. The full model of age, gender, hydroxyurea, AHI, O_2_, PLMI, income, SOL and wake time predicted total sleep time (CSHQ) at V1 and was statistically significant, *F* (9, 230) = 30.53, *p* < 0.001, adjusted *R^2^* = 0.54 (see, Table 7). The model found that TST tends to decrease with: (a) older age, (b) with the use of hydroxyurea and (c) longer sleep onset latency. TST tends to increase with (a) less periodic limb movement, (b) more financial security and (c) later waking time in the morning.

## 4. Discussion

### 4.1. Sleep Profiles: Differences between Polysomnography and Parent Reports

This is the first study to explore the complexity of sleep disorders and sleep characteristics in the young population with SCA using longitudinal approach. Our research has shown that individuals with SCA experience shorter total sleep time (TST) as they grow older, which is in line with previous research shown in the typically developing populations [47,48], including those of similar ethnic background [26]. These findings find support in both sleep measures applied in this study, while only for PSG is an interaction between age difference and TST shown. Sleep onset and offset in PSG is marked with changes in electrical activity of the brain and thus TST recording measures can differ due to subjective understanding of TST in CSHQ (i.e., child is lying in bed, but not actually asleep). PSG is the gold standard of sleep assessment, while subjective parent-reported CSHQ data should be interpreted with caution due to low construct validity with PSG [49]. However, PSG appears as a more reliable marker of developmental, thus biological sleep changes [47,50,51], while CSHQ might be a better marker of behavioural sleep changes, thus making both measures crucial to gain a comprehensive picture of the developmental sleep profiles. Further discrepancies for sleep latency (SL), bedtime and waketime were observed between both measures. Hospital PSG is performed under controlled conditions, which aims to comply with the individual’s usual bedtime and waketime routines at home. However, it is possible that some of the young children might feel anxious having an overnight sleep study in an unfamiliar setting, which can impact on their usual sleep behaviour so that duration is insufficient, or they experience an ‘unusual’ night of sleep [52]. This may explain some of our findings, e.g., that younger children tend to go to bed later during their PSG night. On the other hand, young adults had earlier bedtimes during the PSG night. One explanation for the differences observed might be that individuals feel more familiar with the overnight PSG due to frequent hospital visits or might suggest that they are more tired because of the reduced TST and longer SL as indicated by CSHQ, thus falling asleep more quickly during the PSG night where there are fewer external distractions (e.g., interactions with family, friends and use of technology). The link between increased use of technology at night and lower sleep quantity and quality was observed in 8–17-year-old individuals. However, technology might have been used to help some individuals with sleeping problems to fall asleep [53]. Additional factors contributing to sleep onset delay in preschool and school-aged children in the general population include environmental settings, siblings, health conditions (e.g., allergies and chronic illness), naps and inactivity during the day [54,55,56]. Gender differences observed might be attributed to parental and cultural practice, as well as implication of disease severity on health. For example, males with SCA aged 3 months–11 years experienced more pain crises than females per year [57], which could contribute to the shorter sleep duration observed in our study. Research in a cohort of Chinese children (6–12 years) identified that the strongest predictor for bedtime and waketime in children were school start time, media use, maternal bedtime and waketime, extracurricular activities (e.g., delaying bedtime) and socioeconomic status (e.g., more homework delaying bedtime) [58]. Families of black heritage report having lower parental confidence in managing their child’s sleep compared to families of white heritage, contributing to an average of 27 min later bedtime [59]. Additional, napping behaviour could also delay bedtime and waketime. A significant proportion (39.1%) of children of black heritage still napped at the age of 8 years compared to children from white heritage (4.9%) [24]. In SCA, several unpublished reports noted that napping is also common since additional health conditions (i.e., sleep apnoea and severity of anaemia) increase the risk of daytime sleepiness and fatigue. The continuous confrontation with health problems and the interplay of parenting and socioeconomic factors in SCA from an early age might contribute to the persistence of sleep disturbances across the lifespan, but the relationship needs to be further explored.

### 4.2. Sleep Characteristics

This current study has shown existing sleep disturbances in SCA. In line with our findings, parasomnia was also found to be highest in younger children with SCA (2–6 years, 32%) [13]. Even though our cohort experience sleep disturbances, these seem to be lower compared to the existing literature. We found a 3.9-point lower CSHQ CS (17.7) in our preschool children with SCA, compared to Downes (2016) (21.6) and 23–27% of our cohort habitually snored compared to 54% of 2–17-year-old individuals with SCA in Hankins et al. (2014). More frequent night waking in 4–10-year-old children with SCA was also reported in Daniel et al. (2010) compared to healthy controls. Screening for habitual snoring and night waking is important for this population, since disruption of sleep lowers sleep quality, and lower oxygen levels at night (due to sleep-disordered breathing) increase the risk of pain episodes the next day [19]. Therefore, even fewer symptoms might contribute to persistence of health problems and need to be addressed. However, the majority of preschool children sleep less than the recommended sleep duration by the National Sleep Foundation [46], while about a third of adolescents and young adults seem to sleep more than the recommended amount. One factor that might explain lower sleep quantity and night waking in preschool children is nocturnal enuresis, which occurs in about 70% of 4–5-year-old children [60] and 59% of preschool children with SCA, who wet the bed at least 1–2 time a week [61]. Lehmann and colleagues (2012) showed in this cohort that individuals with SCA (4–19 years) have an increased risk of nocturnal enuresis with co-occurring symptoms of obstructive sleep apnoea [18]. With pubertal onset, teens and young adults experience a sleep phase delay, which, in addition to early school start times, reduces total sleep time [62]. We have also seen a sleep phase delay in our cohort. The midpoint of sleep seems to shift further away from midnight and is especially furthest in older age. The change into adulthood comes with new life changing experiences and the need to be independent, creating the possibility of establishing new sleep habits, which could explain the increase seen in total sleep time for young adults in our study. This finds support in research by Maslowsky and Ozer (2014), who found that African-American adolescents slept longer compared to white and Hispanic individuals. They suggested that longer sleep duration in young adults (19–22 years) occurs because of the freedom to sleep later after transitioning out of high school, but at the same time note that sleep duration decreases once responsibilities increase (i.e., marriage, work and parenthood) [63]. Additional research identified that individuals of black heritage (18–85 years) sleep either too long or too little and that this is attributed to health issues such as heart disease, arthritis and depression [63]. These associations are found in SCA as well [14,64,65]. Currently, only a couple of cross-sectional studies have compared SCA to ethnicity matched controls and found individuals with SCA to have lower TST [36,37,38]. However, the relationships between sleep, sickle cell and health issues need further investigation.

### 4.3. Daytime Functioning

In general, shorter sleep duration is associated with behavioural problems, lower scores on the executive functioning tasks and lower school performance in school-age children [30]. The majority of the SCA population in our study suffered from daytime sleepiness and/or overtiredness. The observed relationship with high CSHQ scores (i.e., parasomnias, SDB and movement at night) and daytime sleepiness is suggestive of impairment in daytime functioning. Recent research in adolescents and young adults with SCA has shown that those with higher sleep fragmentation score (i.e., movement at night as measured with actigraphy) had worse performance on memory recall [66]. Increased movements at night disrupting sleep are indicative of SDB, periodic limb movement (PLM) and parasomnias. Our research has identified that 37% of school aged children experience mild symptoms of PLM, which is much higher compared to healthy children (8%) [67], but similar to a sample of 2–18-year-old children and adolescents with SCA (23.4%) [17]. The high occurrence of SDB and night waking in our sample further contributes to nocturnal sleep disturbances. Children with SCA and sleep-disordered breathing (SDB) may experience processing speed difficulties [6], a cognitive domain important to responding quickly and processing information in one’s environment. Additionally, nocturnal oxygen desaturation, experienced during episodes of SDB, are associated with worse executive function in children with SCA [41]. Fatigue, experienced due to lower oxygen saturation and haemoglobin was associated with lower working memory performance in children with SCA [68]. These cognitive domains are crucial for daytime behaviour. The population with SCA is vulnerable to sleep fragmentation and lower sleep quality early on in life, increasing the risk for impaired daytime and cognitive functioning.

### 4.4. Predictors of Total Sleep Time

Compensatory mechanisms for lack of sleep and daytime sleepiness include an increase in napping behaviour during the day and longer sleeping hours, which aim to ‘catch up’ on lost sleep [69]. Our research has shown that teens and young adults seem to have an increased sleep need, which is contrary to sleep need in healthy adolescence [70]. For example, we found that the amount of sleep tends to decrease in SCA with older age, higher mean overnight O_2_, increase in periodic limb movement, longer sleep onset latency and longer periods of wakefulness during the night. We did not find a relationship with financial security or the use of hydroxyurea for the amount slept during an overnight PSG, but these were associated with CSHQ TST. This suggests that socioeconomic factors play less of a role in a hospital setting, since they follow a structured protocol, which might not reflect bedtime routines at home (i.e., child compliance and parenting). Financial security is an important factor and worry about life and disease can cause anxiety. Recent research has shown that adults with SCA who experience anxiety also experience greater sleep disturbances [71]. The use of hydroxyurea may improve SDB and therefore it may help with disease severity and sleep problems, reducing the need to sleep longer than necessary. The same association is suggested for overnight oxygen saturation, but these hypothetical assumptions needs further careful exploration.

### 4.5. Limitations/Future Directions

Our research was limited to only one overnight PSG per visit. Use of two PSG nights and/or training to familiarise with the new environment might have reduced the differences observed between PSG and CSHQ measures. Recent research has shown that previous training and a suitable schedule can improve child and caregiver experience [72]. It is also important to keep in mind that we did not have ethnicity matched controls and that some participants did not return to their second PSG. Those who did not return at V2 had fewer sleep disorder-related symptoms at V1, which could have had an impacted on our results. Therefore, more longitudinal studies are needed with additional ethnicity matched controls to get a better understanding of the sleep behaviour and profile in SCA. The development of new objective measures such as actigraphy and home PSG might provide a better insight into participants’ nocturnal activity in their habitual environment, since they are less costly and easier to wear over a longer period of time. Future studies should use a combination of measures such as actigraphy with light exposure and questionnaires that collect information on work/school free sleep and work/school sleep times. This can provide important information on individual sleep profiles and sleep needs as well as for the delivery of best sleep practices.

## 5. Conclusions

Sleep disorders and disturbances are present early on in SCA populations. We know that sleep loss can have a significant impact on a young person’s life, impacting positively or negatively on health and is of particular importance in chronic conditions such as SCA. It is important to intervene early on in development to alleviate the course of the disease and to improve health outcomes and quality of life. Sleep serves an important role for daytime and cognitive functioning; hence, quantitative and qualitative assessments of sleep should be mutually considered. Both measures are equally important to guide interventions and can give useful insight for habitual sleep behaviour (i.e., sleep diary, parent and self-report on daily sleep behaviour) or clinical symptoms (i.e., as measured with PSG to diagnose sleep disorders) which would otherwise be missed.

## Figures and Tables

**Figure 1 brainsci-10-00981-f001:**
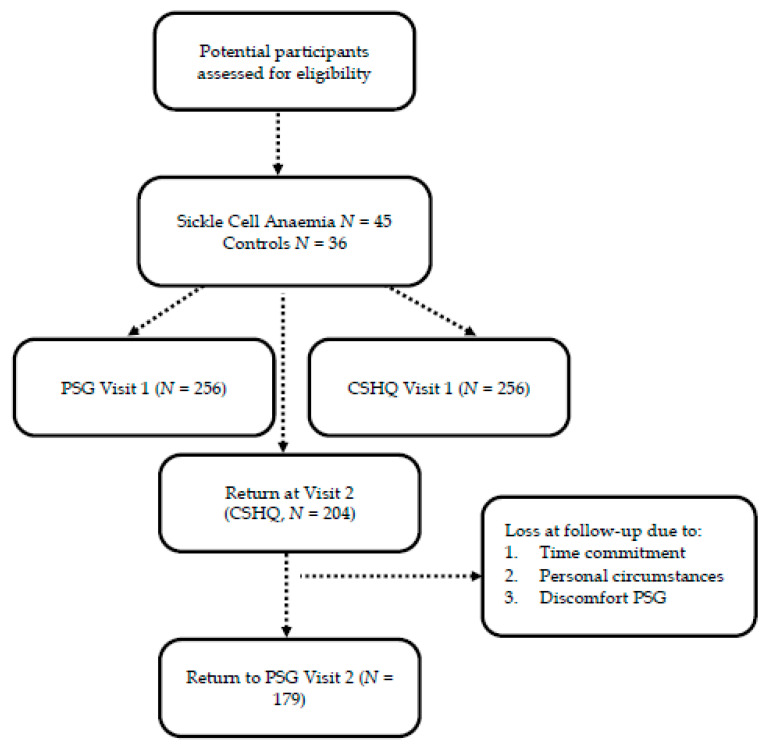
Flowchart. Note. CSHQ = Children’s Sleep Habit Questionnaire, PSG = Polysomnography.

**Figure 2 brainsci-10-00981-f002:**
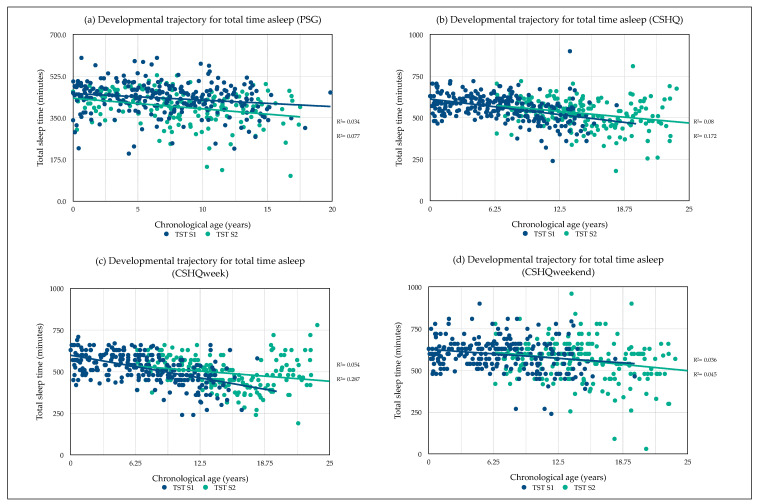
Developmental trajectory for sleep behavirur in sickle cell anaemia. Note: CSHQ = Children’s Sleep Habit Questionnaire, PSG = Polysomnography.

**Table 1 brainsci-10-00981-t001:** Demographics.

	Visit 1	Visit 2	No 2nd Visit	V1-V2	V2-No 2nd Visit
	(*N*), % or Mean ± SD			*p*	
Age in years	11 ± 4.5 (255)	14.9 ± 4.4 (179)	14.9 ± 5.2 (26)	*p* < 0.001	*p* > 0.05
Gender male	51%	50%	58%	*p* > 0.05	*p* > 0.05
Hemoglobin	8.3 ± 1.3 (252)	N/A	N/A	N/A	N/A
Hydroxyurea	12.5% (241)	14.4% (167)	11% (8)	*p* > 0.05	*p* > 0.05
Genotype	(257)	(179)	(77)	*p* > 0.05	*p* < 0.001
SS	90.3%	95.60%	68.8%		
Beta 0 Thalassemia	3.9%	2%	13.0%		
SC	5.4%	2.4%	18.2%		
Black Heritage	(257)	(179)	(78)	*p* > 0.05	*p* > 0.05
UK	36.2%	36.0%	37%		
USA	63.8%	64.0%	63%		
Primary Caregiver	(257)	(179)	(78)	*p* > 0.05	*p* > 0.05
Mother	83.7%	86.0%	78%		
Father	7%	5.6%	10%		
Other	9.3%	8.4%	11%		
Marital Status	(257)	(179)	(78)	*p* < 0.001	*p* < 0.001
Married	15.2%	11.7%	23.0%		
Single	36.2%	31.3%	47.4%		
Living with Partner	5.1%	3.9%	7.7%		
Divorced/Separated	35%	44.7%	12.8%		
SES: Income	(257)	(154)	(64)	*p* > 0.05	*p* > 0.05
Upper middle class	3.9%	3.90%	6%		
Middle class	16.3%	19.50%	19%		
Low income	33.5%	37.00%	39%		
Near poverty level	31.1%	37.00%	36%		
SES: Education	(257)	(176)	(76)	*p* > 0.05	*p* > 0.05
Academic Degree	19.8%	17.6%	26.3%		
Highschool/College	65.8%	70.0%	60.5%		
Primary/Secondary	12.5%	12.5%	13.0%		
Sleep PSG	(256)	(179)	(77)		
AHI	2.5 ± 4.7	2.7 ± 4.5	2 ± 5.2	*p* = 0.021	*p* > 0.05
OAHI	2 ± 4.5	1.5 ± 2.7	1.5 ± 4.5	*p* > 0.05	*p* > 0.05
O_2_	95.7 ± 3.6	95 ± 3.8	96.5 ± 2.9	*p* = 0.015	*p* = 0.001
TST PSG	7 h 13 min	6 h 34 min	6 h 53 min	*p* < 0.001	*p* = 0.043

Note. AHI = Apnea hypopnea index, OAHI = Obstructive apnea hypopnea index, PSG = Polysomnography, SC = Second most common and less severe form of sickle cell anaemia, SES = Socioeconomic status, SS = Most common and severe form of sickle cell anaemia, TST = Total sleep time. Ethnic origin: all from black heritage except one Lebanese. Everyone was fluent in English.

**Table 2 brainsci-10-00981-t002:** Mean ± SD, range and △ Mean ± SD for Child Sleep Habit Questionnaire and Polysomnography for study 1 and 2.

	*N*	CSHQ V1	CSHQ V2	CSHQ V2	*N*	PSG V1	PSG V2	PSG V2	*N*	CSHQ V1	*N*	CSHQ V2
				CSHQ V1				PSG V1		PSG V1		PSG V2
				△ Mean ± SD				△ Mean ± SD		△ Mean ± SD		△ Mean ± SD
Week+ Weekend												
Bedtime	196	21:55 (01:13)	22:51 (01:27)	00:56 (01:17) **	170	22:11 (02:01)	22:24 (00:45)	00:13 (02:01) **	248	–00:15 (01:47) *	170	00:27 (01:27) *
		19:00–03:00	20:00–03:30			20:05–21:51	20:06–00:22					
Sleep Latency	195	36.7 (30.0)	39.8 (33.1)	3.1 (38.9)	170	32.9 (36.4)	26.5 (31.9)	−6.4 (47.1) *	248	5.91 (46.0) *	169	14.2 (43.2) **
		0–180	0–180			0–235	0–181					
Waketime	197	08:05 (00:57)	08:33 (1:23)	00:28 (01:16) **	170	06:57 (00:37)	06:29 (0:22)	–00:28 (00:42) **	250	01:09 (01:09) **	170	02:04 (01:30) **
		05:45–11:00	05:45–14:00			02:44–08:17	4:59–07:18					
Hours per day	197	09:23 (01:189	08:47 (01:26)	–00:36 (01:38) **	170	07:20 (01:05)	06:34 (01:11)	–00:46 (01:25) **	249	02:07 (01:36) **	170	02:11 (01:40)**
		04:00–15:00	03:00–13:30			03:42–10:02	01:46–08: 49					
Week												
Bedtime	200	21:19 (01:13)	22:04 (01:25)	00:45 (01:18) **	50	22:14 (01:00)	22:37 (00:50)	00:23 (01:01) *	80	–01:08 (01:22) **	101	–00:17 (01:33) *
		19:00–03:00	19:30–03:00			20:50–01:11	20:06–00:22					
Sleep Latency	198	39.4 (32.7)	40.7 (37.92)	1.3 (42.1)	50	27.5 (21.9)	32.1 (35.7)	4.6 (43.1)	80	6.9 (44.0) *	101	15.6 (49.0) *
		0–180	0–180			0–117	0–150					
Waketime	199	06:59 (00:53)	07:18 (01:26)	00:18 (01:32)	50	06:47 (00:21)	06:27 (00:25)	–00:20 (00:34) **	80	00:23 (00:55) *	101	01:00 (01:35) **
		04:00–11:00	05:00–15:00			05:3–07:38	04:59–07:14					
Hours per day	199	08:46 (01:29)	08:13 (01:33)	–00:33 (01:48) **	50	7:03 (00:58)	06:12 (01:14)	–00:51 (01:01) **	80	02:12 (01:41) **	101	01:48 (01:41) **
		04:00–11:49	03:10–13:00			04:49–08:55	02:10–08:19					
Weekend												
Bedtime	197	22:32 (01:28)	23:39 (01:46)	01:06 (01:45) **	120	22:09 (02:19)	22:19(00:42)	00:09 (02:18) *	170	00:29 (02:08) **	69	01:44 (01:54) **
		19:00–04:00	20:00–05:00			20:05–21:51	20:18–00:16					
Sleep Latency	197	33.7 (35.3)	38.4 (39.5)	4.7 (49.3)	120	35.1 (40.9)	24.1 (30.0)	–11.0 (48.1) *	170	7.0 (55.0)	69	–7.1 (43.6)
		0–180	0–240			0–235	0–181					
Waketime	200	09:12 (01:35)	09:50 (02:00)	00:38 (01:51) **	120	07:01 (00:41)	06:30 (00:21)	–00:31 (00:44) **	174	02:11 (01:38) **	69	03:18 (02:13) **
		05:30–14:00	05:00–15:00			02:44–08:17	05:29–07:18					
Hours per day	198	10:00 (01:41)	09:21 (02:01)	–00:38 (02:19) **	120	07:27 (01:07)	06:43 (01:08)	–00:43 (01:21) **	171	02:25 (01:50) **	66	03:03 (01:55) **
		04:00–19:00	00:30–16:00			03:42–10:02	01:46–08:49					

*Note.* CSHQ = Children’s Sleep Habits Questionnaire, PSG = Polysomnography. All *p* values for Wilcoxon’s signed rank test. (* *p* < 0.5 and ** *p* < 0.01).

**Table 3 brainsci-10-00981-t003:** Mean (±SD) and range for PSG Scores at visit 1 and 2.

	*N*	PSG Visit 1	PSG Visit 2	PSG V2-PSG V1	*p*
				△ Mean (±SD)	
AHI	170	2.7 (4.5)	1.9 (2.8)	0.8 (4.2)	*p* = 0.021
		0–37.4	0–19.6		
WASO	170	59.1 (41.9)	63.7 (53.2)	−4.7 (62.5)	*p* > 0.05
		9–277.5	5.5–378		
Night waking	170	25.4 (8.2)	26.4 (9.6)	−1 (10.8)	*p* > 0.05
		8–74	8–61		
PLM	169	5.9 (10.5)	4.7 (8.2)	1.2 (9.8)	*p* = 0.002
		0–117.3	0–72.7		
O_2_ Saturation	170	95.3 (3.8)	94.9 (3.8)	0.4 (2.8)	*p* = 0.015
		81.2–100	80.5–100		

Note. AHI = Apnea and Hypopnea Index, O_2_ = Oxygen, PLM= Periodic leg movement, PSG = Polysomnography, WASO = Wake after sleep onset. All *p* values for Wilkoxon’s signed rank test.

**Table 4 brainsci-10-00981-t004:** Mean (±SD) and range for CSHQ Scores at visit 1 and 2.

	*N*	CSHQ Visit 1	CSHQ Visit 2	CSHQ V2-CSHQ V1	*p*
				△ Mean (±SD)	
CSHQ CS	200	14.9 (10.1)	13.5 (9.7)	1.4 (10.7)	*p* = 0.037
		0–49	0–55		
SDB	200	3.0 (3.1)	3.1 (3.0)	−0.02 (3.4)	*p* > 0.05
		0–16	0–16		
Parasomnias	200	5.4 (4.3)	4.6 (3.8)	0.8 (4.3)	*p* = 0.019
		0–21	0–23		
Night wake	200	3.5 (3.5)	2.7 (3.2)	0.7 (4.2)	*p* = 0.01
		0–12	0–12		
Bedtime resistance	200	1.2 (1.5)	1.0 (1.5)	0.2 (1.8)	*p* > 0.05
		0–4	0–4		

Note. CSHQ = Children’s Sleep Habits Questionnaire, CS = Composite score. All *p* values for Wilkoxon’s signed rank test.

**Table 5 brainsci-10-00981-t005:** Mean (±SD) for daytime sleepiness and CSHQ scores.

Daytime Sleepiness
*N*(CSHQ Score Mean ± SD)	Never	Not Often	Sometimes	Often	Always	*p*
CSHQ CS Visit 1	162(12.2 ± 8.6)	35(16.1 ± 9.9)	38 (18.1 ± 11.3)	10 (23.3 ± 7.5)	10 (25.4 ± 9.8)	*p* < 0.001
CSHQ CS Visit 2	102 (9.1 ± 6.6)	23 (12.5 ± 5.8)	47 (16.7 ± 8.8)	14 (23.2 ± 13.6)	15 (27.1 ± 9.2)	*p* < 0.001
SDB Visit 1	162 (2.5 ± 2.8)	35 (3.1 ± 3.1)	38 (4 ± 3.5)	10 (6.1 ± 3.2)	10 (4.6 ± 3.1)	*p* < 0.001
SDB Visit 2	102 (2.4 ± 2.4)	23 (2.6 ± 1.7)	47(3.6 ± 3.5)	14 (5 ± 3.9)	15(4.7 ± 3.9)	*p* < 0.05
Movement at night Visit 1	162 (0.6 ± 1.3)	35 (1.4 ± 1.7)	38 (1.7 ± 2)	10 (2.4 ± 1.8)	10 (1.5 ± 1.8)	*p* < 0.001
Movement at night Visit 2	102 (0.5 ± 1.2)	23 (0.7 ± 1.3)	47 (1.4 ± 1.7)	14 (2.1 ± 2.7)	15 (2.7 ± 2.4)	*p* < 0.001
Night wake Visit 1	162 (3 ± 3.3)	35 (3.3 ± 3.1)	38 (3.4 ± 3.1)	10 (3 ± 3.2)	10 (7.3 ± 4.5)	*p* < 0.001
Night wake Visit 2	102 (1.7 ± 2.6)	23 (2.7 ± 2.7)	47 (3.7 ± 2.9)	14 (3.9 ± 4.1)	15 (5.9 ± 4.8)	*p* < 0.001
Parasomnias Visit 1	162 (4.4 ± 4)	35 (5.7 ± 4.4)	38 (6.4 ± 4.4)	10 (8.5 ± 4)	10 (8.9 ± 5.9)	*p* < 0.05
Parasomnias Visit 2	102 (3.2 ± 3.2)	23 (4.1 ± 2)	47 (5.6 ± 3.6)	14 (8 ± 4.8)	15 (9.5 ± 4.6)	*p* < 0.001
Napping Visit 1	161 (0.8 ± 1.2)	35 (1.2 ± 1.3)	38 (1.2 ± 1.3)	10 (1.8 ± 1.5)	10(2.4 ± 1.8)	*p* < 0.05
Napping Visit 2	103 (0.9 ± 1.3)	23 (1 ± 1.1)	48 (1.7 ± 1.4)	14 (2.5 ± 1)	15 (1.5 ± 1.8)	*p* < 0.001
TST Visit 1	159 (9:18 ± 1:21)	35 (9:11 ± 1:18)	38 (9:31 ± 1:04)	10 (9:23 ± 1:23)	8 (9:59 ± 1:17)	*p* > 0.05
TST Visit 2	100 (8:49 ± 1:23)	23 (8:44 ± 1:09)	48 (8:50 ± 1:33)	14 (8:58 ± 1:27)	15 (8:13 ± 2:03)	*p* > 0.05
Haemoglobin Visit 1	159 (8.3 ± 1.4)	34 (8.4 ± 1)	38 (8.4 ± 1.4)	10 (8.4 ± 1)	9 (7.9 ± 1.5)	*p* > 0.05
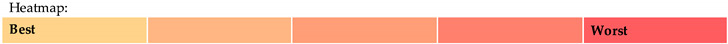

*Note.* CSHQ = Children’s Sleep Habits Questionnaire, SDB = sleep disordered breathing. Not often (< 1 day a week), Sometimes (1–2 days a week), Often (3–5 days a week) and Always (6–7 days a week). TST = Total sleep time. *All p* values for Kruskal-Wallis test.

**Table 6 brainsci-10-00981-t006:** Hierarchical multiple regression predicting total sleep time (PSG) at V1.

		Total Sleep Time		
Model	Variable	β	*p*-Value	R^2^
Step 1	Age	−0.17	0.007	0.03 *
Step 2	Age	−0.19	0.003	0.06 *
	O_2_	0.06	0.365	
	PLMI	−0.16	0.013	
Step 3	Age	−0.23	<0.001	0.54 **
	O_2_	−0.14	0.002	
	PLMI	−0.11	0.027	
	SOL	0.3	<0.001	
	WASO	−0.48	<0.001	
	Night waking	0.16	0.001	
	Waketime	0.41	<0.001	

Note. PLMI = Periodic leg movement index, SOL = Sleep onset latency, O_2_ = Mean overnight oxygen saturation, WASO= Wake after sleep onset. *N* = 230. * *p* < 0.05, ** *p* < 0.001.

**Table 7 brainsci-10-00981-t007:** Hierarchical multiple regression predicting total sleep time (CSHQ) at V1.

		Total Sleep Time		
Model	Variable	β	*p*-Value	R^2^
Step 1	Age	−0.46	<0.001	0.22 **
	Female	0.05	0.349	
Step 2	Age	−0.45	<0.001	0.25 **
	Female	0.06	0.319	
	Hydroxyurea	−0.11	0.058	
	AHI	−0.03	0.682	
	O_2_	−0.03	0.694	
	PLMI	0.15	0.012	
Step 3	Age	−0.43	<0.001	0.27 **
	Female	0.07	0.258	
	Hydroxyurea	−0.12	0.042	
	AHI	−0.03	0.589	
	O_2_	−0.03	0.603	
	PLMI	0.15	0.009	
	SES	0.13	0.024	
Step 4	Age	−0.49	<0.001	0.55 **
	Female	0.04	0.403	
	Hydroxyurea	−0.09	0.048	
	AHI	−0.06	0.229	
	O_2_	−0.07	0.182	
	PLMI	0.12	0.007	
	SES	0.11	0.022	
	SOL	−0.35	<0.001	
	Waketime	0.45	<0.001	

Note. AHI = Apnoea and hypopnoea index, PLMI = Periodic leg movement index, SES = Socioeconomic status, SOL = Sleep onset latency, O_2_ = Mean overnight oxygen saturation. *N*= 231. ** *p* < 0.001.

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
