# Peer review of "Developmental Profile of Sleep and Its Potential Impact on Daytime Functioning from Childhood to Adulthood in Sickle Cell Anaemia"

_brainsci, 2020, doi:10.3390/brainsci10120981_

Round 1

Reviewer 1 Report

Manuscript ID: brainsci-983332-peer-review-v1

Title: Developmental profile of sleep and its potential impact on daytime functioning in children and adolescent with sickle cell anaemia

Journal: Brain Sciences

Abstract

1) Lines 17-18. Authors should briefly describe the procedure of the original study to give the readers the opportunity to fully understand the abstract.

2) Line 24. Authors should replace “included;” with “included:”.

Introduction

1) Line 59 and others. Authors summarized the findings of several studies not specifically carried out on patients with sickle cell anaemia. My suggestion is that authors only review previous studies on sleep in sickle cell anaemia patients.

2) Lines 82-83. Authors wrote “There are cross-sectional studies involving questionnaires, but few have compared with contemporaneous PSG or included control groups and/or gathered longitudinal data”. In line with my previous comment, authors should summarize the findings of these studies.

3) Line 88. Authors should also examine the longitudinal variation in sleep efficiency, marker of sleep quality, and midpoint of sleep, marker of sleep timing.

Materials and methods

1) Line 92. Authors should report the details for ethical approval.

2) Line 94. Authors should replace “Each child” with “Each child and adolescent”.

3) Line 102. In this section authors actually presented the tools used to assess sleep but missed to describe the procedure. Authors are strongly invited to describe in details the procedure.

4) Line 127. Authors should include in the statistical analyses only participants with completed data (both questionnaires and PSG) both at visit 1 and visit 2. Moreover, they should report within the main text only the results of statistical analyses carried out separately for age group.

Results

1) Lines 140-144. Authors should report these data in a flow-chart.

2) Line 150. Table 1 should also show the complete statistics referred to each comparison. Moreover, authors should clarify the origin of sleep PSG data appearing in the column “No 2nd Visit”.

3) Lines 153-155. Authors wrote “Differences were also observed for participants who did not return for their second PSG visit (Table 1). They tend to be less symptomatic compared to those returning for their second PSG night (all p <0.05)”. These differences could point out a reduced representativity of the final sample of patients with available PSG data at visit 2.

4) Line 160. Table 2 should show the complete statistics referred to each comparison.

5) Lines 164-166. Authors present the results separately for age group, but corresponding data are not reported in Table 2.

6) Line 209. Authors should report the complete statistics in Table 3.

7) Line 266. It would be extremely interesting to examine also the predictors of sleep efficiency and midpoint of sleep.

8) Lines 272-273. Authors should explain in details the reasons why those predictors were removed.

Discussion

1) Lines 303-304. Reading the introduction, it seemed that few longitudinal studies were already carried out.

2) Line 386. Authors should replace “Actiwatch” with “Actiwatch actigraph”.

Author Response

Dear Reviewer,
Thank you very much for taking the time to review our research and especially thank you for
your kind and helpful suggestions to our work. We made the changes visible in red underlined
in the new document brainsci-983332. Please find our reply to your questions below.
-------------------------------
Abstract
1) Lines 17-18. Authors should briefly describe the procedure of the original study to give the readers the opportunity to fully understand the abstract.
2) Line 24. Authors should replace “included;” with “included:”.
Regarding to your two comments, we have added a sentence in the Abstract (line 16–17) and in the Method section (line 124-127). Semicolon was changed to a colon (line 25).
-------------------------------
Introduction
1) Line 59 and others. Authors summarized the findings of several studies not specifically carried out on patients with sickle cell anaemia. My suggestion is that authors only review previous studies on sleep in sickle cell anaemia patients.
Regarding this paragraph, it was our intention to show that subjective sleep data
collected by parents is different to a child’s objective sleep data. Unfortunately,
research solely in individuals from black heritage is sparse and we need to refer back to the general population from white or other heritage. This research is not available in individuals from black heritage to our current knowledge and since our research aims to show the difference between objective and subjective sleep measures as well, we believe that it is important to have this research added in the introduction. And we do mention this in our introduction (line 67 – 69) that the “[…] majority of studies include relatively small cohorts from different racial backgrounds, with the literature […] missing sleep characteristics data in young children of black heritage […].”
However, we did add more research on sickle cell participants and sleep in the
introduction (line 91-99).
2) Lines 82-83. Authors wrote “There are cross-sectional studies involving questionnaires, but few have compared with contemporaneous PSG or included control groups and/or gathered longitudinal data”. In line with my previous comment, authors should summarize the findings of these studies.
Thank you very much again for your suggestion. Currently there is no longitudinal
research available about the sleep profile in sickle cell anaemia and we made this clear now in the introduction (line 91). As per your suggestion to your question 1 above, we have added more research to the introduction to better explain the missing research and our rationale for the study (line 91-99).
3) Line 88. Authors should also examine the longitudinal variation in sleep efficiency, marker of sleep quality, and midpoint of sleep, marker of sleep timing.
We do agree that this would be very interesting and important and would add to the significance of our sleep research in sickle cell anaemia and sleep. The aims of our research were to understand the overall sleep profile and behaviour in sickle cell anaemia, since they show a high occurrence of sleep disorders without the focus on circadian rhythmicity. Additionally, we wanted to show that there is a difference in the data collected, depending on which instrument is used and unfortunately, we only have data available on sleep efficiency with PSG. Certainly, actigraphy would be very helpful for future studies to gather more ecological valid sleep information on actual total sleep time and sleep quality in this population to get a better understanding on their daily sleep behaviour.
When the study was designed and conducted no information on chronotype, exposure to outdoor light, sleep as on work/ school free days and on work/ school days was collected, such as with the Munich Chrono Type Questionnaire (MCTQ). We collected information on sleep behaviour at week and weekend during the last month and these are not enough to gain an ecologically valid understanding of their mid time of sleep.
For example, many of our older participants worked during the weekend, studied or had important hospital appointments and therefore we are only able to give an overall midpoint of sleep without in depth interpretation on chronotypes and accumulation of sleep depth.
We thank you again for stressing the importance of sleep quality and midpoint of sleep.
Therefore, we added information on sleep efficiency (line 219-220) and midpoint of sleep to our work (line 269-273) and mentioned it in methods (line 153-154)
introduction (line 82-85), discussion (line 494-495) and the limitation section (line
562-565). We hope that there will be future studies to discuss this important issue and to guide interventions.
-------------------------------
Materials and Methods
1) Line 92. Authors should report the details for ethical approval.
We have added our ethical approval (line 108-112)
2) Line 94. Authors should replace “Each child” with “Each child and adolescent”.
Thank you. We have checked the MS and now corrected throughout.
3) Line 102. In this section authors actually presented the tools used to assess sleep but missed to describe the procedure. Authors are strongly invited to describe in details the procedure.
Thank you very much again for your advice. We made the changes to question 2 (line 112-113) and question 3 (line 124-127).
4) Line 127. Authors should include in the statistical analyses only participants with completed data (both questionnaires and PSG) both at visit 1 and visit 2. Moreover, they should report within the main text only the results of statistical analyses carried out separately for age group.
Thank you again for your kind suggestions. We can say with certainty that we have
analysed only data of participants with a complete data set. Referring back to table 2, which we have now clarified (7 of 22). Our apologies if it was misleading, but we have used all the data that was available to show the mean (±SD) for each visit separately to gain an overall understanding of sleep behaviour and then used only data from participants that was available at visit 1 and 2.
It was our aim to show the differences in sleep profiles between visit 1 and 2 for data collected form PSG and CSHQ. To investigate if there are differences between the age groups we added an additional analysis. Due to the large amount of information for this comparison we have made it available in the supplementary material (see supplementary table 3-4) and given the main results in the text. We have added the information on the analysis used in ‘2.3. Statistical analysis’ (line 160-161).
However, if you feel that these tables add more value and should be in the main text, we are happy to move these from the supplementary material to the main text.
-------------------------------
Results
1) Lines 140-144. Authors should report these data in a flow-chart.
Thank you again for your suggestion. We have added now a flow chart to the main text (line 181).
2) Line 150. Table 1 should also show the complete statistics referred to each comparison.
Moreover, authors should clarify the origin of sleep PSG data appearing in the column “No 2nd Visit”.
We have added all the p-values to table 1 (page 6 of 22) and clarified what we mean by 2nd Visit in the main text (line 200-201).
3) Lines 153-155. Authors wrote “Differences were also observed for participants who did not return for their second PSG visit (Table 1). They tend to be less symptomatic compared to those returning for their second PSG night (all p <0.05)”. These differences could point out a reduced representativity of the final sample of patients with available PSG data at visit 2.
Thank you very much for your suggestion. Yes, we do agree that this could reduce the representative of the sample and we have added this to our limitation section (line 555-560). However, since the sample seems to be less symptomatic we argue that they could have fewer sleep related problems and therefore might fall into the minority of the SCA group that are doing better, but we cannot tell with certainty.
4) Line 160. Table 2 should show the complete statistics referred to each comparison.
We have combined table 2 with supplementary table 2 to show the complete statistics (page 7 of 22).
5) Lines 164-166. Authors present the results separately for age group, but corresponding data are not reported in Table 2.
All the data for the different age groups can be found in Supplementary table 2 and 3.
We have not added the whole tables in the text, because of the detailed amount of data available, however, this comprehensive data is available in the Supplementary Material.
We made sure that the description, where to find the data occurs at the beginning of the text, and before data description (line 206-208).
6) Line 209. Authors should report the complete statistics in Table 3.
We have improved table 3 and 4 with the complete statistics (page 9 and 10 of 22).
7) Line 266. It would be extremely interesting to examine also the predictors of sleep efficiency and midpoint of sleep.
We agree that this would be extremely interesting to do. However, the main aim of the study was it to look at the developmental trajectory of sleep in sickle cell anaemia and to compare different assessment measures, therefore we have chosen only to look at total sleep time, since this was available in PSG (i.e. giving an insight into sleep disorders and impact on total sleep time) and CSHQ (i.e. giving an indication of daily sleep behaviour and the potential impact sleep disorders have on the daily life) and we do not feel strongly to add an additional
regression analysis to this manuscript. However, we have added information on sleep efficiency and midpoint of sleep. We are currently working on another project that includes actigraphy and cortisol data and we will certainly make sure to include sleep efficiency and midpoint of sleep here, since the actigraphy is a wonderful objective measure to better understand our participants daily sleep behaviour. We are happy to send you the publication, if you are interested once it is reviewed and accepted.
8) Lines 272-273. Authors should explain in details the reasons why those predictors were removed.
None of the variables had an influence on the full model. We have added it to the main text (line 384-385).
-------------------------------
Discussion
1) Lines 303-304. Reading the introduction, it seemed that few longitudinal studies were already carried out.
Thank you again for pointing out that it seems that few longitudinal studies were
already carried out. Our apology if we did not make it clear. We hope that the added information to the introduction makes it clear now that there are no longitudinal studies available at the moment.
2) Line 386. Authors should replace “Actiwatch” with “Actiwatch actigraph”.
Thank you again for pointing this out. We have now changed it to the commonly used word in the literature: actigraphy.
Thank you very much again for your time and patience and for helping to improve our research in sickle cell anaemia and sleep. We hope that we were able to address your comments and are happy to discuss anything further.

Reviewer 2 Report

Kolbel and colleagues provide an in depth analysis of polysomnography and sleep questionnaires to understand developmental sleep profiles in children and adolescents with sickle cell anemia. My only major concern would be that there is no comparison group to compare the children with SCA to children without SCA. The literature shows that as children age into adults, there is a loss of sleep during normal development so the authors showing that SCA children losing sleep as they age is similar to past work. How do children with SCA differ from children without? I know this was not necessarily their question when performing their study, but a comparison between how SCA children and non-SCA children sleep when measured in the hands of the experimenters would be nice to see. 

There were many abbreviations in the study. In some cases the abbreviation was only used one other time (e.g. wake time (WT)). I would suggest cutting some of these superfluous abbreviations out. 

Author Response

Dear Reviewer,
Thank you very much for taking the time to review our research and especially thank you for your kind and helpful suggestions to our work. We made the changes visible in red underlined in the new document brainsci-983332.
The aim of our study was to show the sleep profile of individuals with sickle cell
anaemia over time, since there is no longitudinal research currently available and at the time of the original study and participant recruitment no funding for control data was available (although it was requested) so data could not be collected.
We have written a review of the data available on sleep in sickle cell anaemia as part of the first authors doctoral thesis which we aim to submit it for review and publication soon. We are happy to send you the reviewed and accepted manuscript, if you are interested.
Koelbel, M., Dimitrou, D. & Kirkham, F. J. (2020). [Systematic Review and Meta-analysis of Sleep Behaviour and Habits in Sickle Cell Anaemia]. Unpublished.
Currently there are only a couple of studies available that have used ethnicity matched control data and that we can compare to. We do agree that this would support and strengthen our work and therefore we added a couple of sentences in our introduction (line 91-99) and discussion (line 505-507), and limitation section (line 555-560). We hope that future research will include ethnicity matched controls to better understand how individuals with sickle cell compare to healthy individuals.
We also made sure to use the abbreviations appropriately.
Thank you very much again for your time and patience and for helping to improve our research in sickle cell anaemia and sleep. We hope that we were able to address your comments and are happy to discuss anything further.

Round 2

Reviewer 1 Report

Manuscript ID: brainsci-983332-peer-review-v2

Title: Developmental profile of sleep and its potential impact on daytime functioning in children and adolescent with sickle cell anaemia

Journal: Brain Sciences

General suggestions

  1. In order to facilitate the review, authors could simply highlight the changes in bold or colored text, without using track changes. Moreover, authors wrote “We made the changes visible in red underlined in the new document brainsci-983332” but I was not able to find parts of the manuscript in red underlined.
  2. Authors should check that the numbers of lines they are referring to within their replies actually correspond to those within the revised manuscript.

Specific comments

  1. Pages 6 and 7. Two different “Table 1” are reported.
  2. Pag 8. Two “Table 2” are reported.
  3. Page 11. Two “Table 3” are reported.
  4. Page 13. Two “Table 4” are reported.

Author Response

Dear Reviewer,

Thank you very much for the suggestions. We have highlighted now the changes in the main manuscript and removed the track changes in the word document.

Only one new table for Table 1,2,3 and 4, revised according to your kind comments, was reported and added to the manuscript. However, the tracked changes showed the removed table on the side. We hope that you are able to see everything now and that it is less confusing without the tracked changes.

Thank you very much again for your time and help.